# Changing risk factors for placental abruption: A case crossover study using routinely collected data from Finland, Malta and Aberdeen

Emma Anderson[1], Edwin Amalraj Raja[2], Ashalatha Shetty[3], Mika Gissler[4,5], Miriam Gatt[6], Siladitya Bhattacharya[7], Sohinee Bhattacharya[8]*

1 University of Aberdeen Medical School, Aberdeen, United Kingdom, 2 Division of Applied Health Sciences, Medical Statistics Team, University of Aberdeen, Aberdeen, United Kingdom, 3 Aberdeen Maternity Hospital, Aberdeen, United Kingdom, 4 Information Services Department, THL National Institute for Health and Welfare, Helsinki, Finland, 5 Department of Neurobiology, Care Sciences and Society, Karolinska Institute, Stockholm, Sweden, 6 Strategy and Sustainability Division, Ministry for Health, Valletta, Malta, 7 School of Medicine, Medical Sciences and Nutrition, University of Aberdeen, Aberdeen, United Kingdom, 8 Dugald Baird Centre for Research on Women's Health, Aberdeen Maternity Hospital, Aberdeen, United Kingdom

* sohinee.bhattacharya@abdn.ac.uk

**Data Availability Statement:** The dataset was created from three population based international datasets and permissions obtained from governing committees for the 3 databases. Therefore

## Abstract

### Objective

To evaluate the effects of changes in risk factors between the first two pregnancies on the occurrence of placental abruption (PA) in the same woman.

### Methods

Routinely collected obstetric data from Aberdeen Maternity and Neonatal Databank, the Maltese National Obstetric Information System and the Finnish Medical Birth Register were aggregated. Records of the first two singleton pregnancies from women who had PA in one pregnancy but not the other, were identified from this pooled dataset. A case-crossover study design was used; cases were pregnancies with abruption and matched controls were pregnancies without abruption in the same woman. Conditional logistic regression was used to investigate changes in risk factors for placental abruption in pregnancies with and without abruption.

### Results

A total of 2,991 women were included in the study. Of these 1,506 (50.4%) had PA in their first pregnancy and 1,485 (49.6%) in a second pregnancy. Pregnancies complicated by pre-eclampsia {194 (6.5%) versus 115 (3.8%) adj OR 1.69; (95% CI 1.23–2.33)}, antepartum haemorrhage of unknown origin {556 (18.6%) versus 69 (2.3%) adjOR 27.05; 95% CI 16.61–44.03)} and placenta praevia {80 (2.7%) versus 21 (0.7%) (adjOR 3.05; 95% CI 1.74–5.36)} were associated with PA. Compared to 20 to 25 years, maternal age of 35–39 years {365 (12.2) versus 323 (10.8) (adjOR 1.32; 95% CI 1.01–1.73) and single marital

permission for public access to data will need to be given by all three committees. The Finnish register data have been given for this specific study, and the data cannot be shared without authorization from the register keepers. More information on the authorization application to researchers who meet the criteria for access to confidential data can be found at https://thl.fi/fi/web/thlfi-en/statistics/ information-for-researchers/authorisation- application (THL). Similarly data from Aberdeen can be accessed by applying to the AMND steering committee found at https://www.abdn.ac.uk/iahs/ research/obsgynae/amnd/access.php. The authors did not have special access privileges in accessing the data.

**Funding:** Funding was received from NHS Grampian Endowment Fund (Grant number RG14524-10) to cover data access and storage costs and for article processing charges for open access publication for this research project. The funders played no role in study design, data collection and analysis, decision to publish, or preparation of the manuscript.

**Competing interests:** Prof Siladitya Bhattacharya receives honorarium and travel expenses from Oxford University Press as Editor in Chief of Human Reproduction Open journal. As invited speaker, has received funding for travel and accommodation at conferences supported by Industry and received research funding from National Institute for Health Research UK. The corresponding author Dr Sohinee Bhattacharya is the wife of Prof Siladitya Bhattacharya. She has received research grants from MRC that is not related to this project in the last 5 years. This does not alter our adherence to PLOS ONE policies on sharing data and materials. All other authors have declared that they have no competing interests.

status (adjOR 1.36; 95% CI 1.04–1.76) were independently associated with PA. Maternal smoking, BMI and fetal gender were not associated with PA.

## Conclusion

Advanced maternal age, pregnancies complicated with unexplained bleeding in pregnancy, placenta praevia and preeclampsia were independently associated with a higher risk of placental abruption.

## Introduction

Placental abruption (PA) is an important cause of antepartum haemorrhage (APH) that affects 0.3–1% of pregnancies. [1] Defined as the premature separation of the placenta from the uterine wall, PA usually occurs without warning between 24 weeks gestation and delivery, [2] and is caused by rupture of the decidual vessels and haemorrhage within the placental bed. [3] Abruption can be revealed, indicated by vaginal bleeding, or concealed, where haemorrhage is contained behind the placenta. [4] The aetiology is unknown, and possibly part of a wider placental syndrome caused by underlying vascular pathology associated with defective deep placentation. [5] Oxygen supply to the fetus is compromised and maternal blood loss may be significant in affected women. Prompt fetal monitoring, maternal hemodynamic stabilization [2] and delivery, commonly by caesarean section (90%), is indicated within 24 hours of abruption. [4, 6] PA may lead to antepartum fetal death and disseminated intravascular coagulopathy, though maternal mortality is rare with good healthcare access [6].

While PA can be triggered by abdominal trauma, most cases are not preceded by a clear pre-disposing event. Sociodemographic risk factors include maternal race/ethnic background, BMI, social class, marital status [7] and extremes of maternal age. [8] Behavioural risk factors include smoking, cocaine use, alcohol and short interpregnancy interval. [7] Smoking is one of the strongest established risk factors and exhibits a dose-response relationship. [9] Diabetes and hypertensive disease such as pre-eclampsia may aggravate the underpinning microvascular dysfunction, thus causing abruption. [10] Vaginal bleeding in pregnancy, placenta praevia and premature rupture of membranes (PROM) are also significant risk factors [7] as are stillbirth or abruption in a previous pregnancy. [2,11] PA shows aggregation within families [12] and has an association with heritable thrombophilias. [13] However, published literature is often inconsistent on the significance and importance of modifying these risk factors.

The objective of this study was to evaluate any changes in risk factors associated with PA across two consecutive pregnancies in the same woman by controlling for woman level variables such as inherited risk between pregnancies.

## Materials and methods

### Study design

A case-crossover study design was used in women who experienced pregnancies with and without PA, such that they acted as their own controls.

### Data sources

This study used anonymised data from three sources—the Aberdeen Maternity and Neonatal Databank (AMND) between 1986 and 2012, the Maltese National Obstetric Information

System (NOIS) between 1999 and 2015 and Finnish Medical Birth Register (MBR) between 1987 and 2014. All three contain routinely collected clinical information on maternal, obstetric and neonatal characteristics of deliveries at or over 22 weeks' gestation. Maltese NOIS and Finnish MBR are national databases collecting data from all maternity hospitals in the country, [14, 15] while AMND collects data on all births within Aberdeen City District—a defined geographical region of Scotland. [16] The pooled dataset comprised women with their first two singleton pregnancies between 1986 and 2015. Women with missing information on placental abruption were excluded. The population selection process is shown in Fig 1.

## Ethical approval

Permissions to analyse anonymised data were obtained from the Caldicott guardians of all three databases: the steering committee of the Aberdeen Maternity and Neonatal Databank (AMND 3/2016); National Institute for Health and Welfare, Finland (THL 1719/5.05.00/ 2015); Directorate for Health Information Research Malta (28/04/2016). As routinely collected anonymised data were analysed formal ethical approval was not considered necessary by the North of Scotland Research Ethics Service. This analysis was part of a collaborative project looking at recurrence risk of stillbirth.

## Case definition

Placental abruption was coded according to the International Classification of Diseases 9[th] or 10[th] Revision (ICD—9/10) in all three databases. ICD 10 defines placental abruption as 'The separation of the placenta from the maternal uterine attachment when it occurs after the twentieth week of the pregnancy.'[17] Since 1 Oct 1990 Finland had a separate check-box for placental abruption. Pregnancies without placental abruption in the same women were the controls. Therefore the cases and control pregnancies were matched within each woman included in the study.

## Risk factors

The potential risk factors under investigation included maternal age category (<20, 20–24, 25–29 [reference],30–34, 35–39 and ≥40 years), parity, BMI category (<18.5 as underweight, 18.5–24.9 as normal [reference], 25–29.9 as overweight and ≥30 as obese), smoking status (yes vs no), deprivation status (Deprived vs not deprived), marital status (single vs married), gestational diabetes, gestational hypertension, pre-eclampsia, threatened miscarriage, antepartum haemorrhage (APH) of unknown origin, placenta praevia, maternal anaemia, premature rupture of membranes (PROM) (yes versus no) and gender of the baby (male vs female). In each source database, variables were checked and re-coded, where necessary, to ensure homogenised coding amongst the three datasets. Continuous variables such as age and BMI were categorised prior to analysis; categorisation and reference bands were based on existing literature. Marital status 'single' denoted single, widowed, divorced, separated and 'Married' denoted marriage or co-habitation. Social class was measured differently between the three source data sets. These were re-coded into a new variable 'Deprivation status' for consistency and data merging. AMND recorded Registrar General's paternal occupation based social class recoded as binary 'not deprived' and 'deprived'. Finland used maternal occupational classification; 'upper white-collar worker' counted as 'not deprived', all others (lower white-collar, blue-collar and other including student and housewife) as deprived. Malta used maternal level of education attained as a proxy for social class. University level education was coded into 'non-deprived' and below this level as 'deprived'.

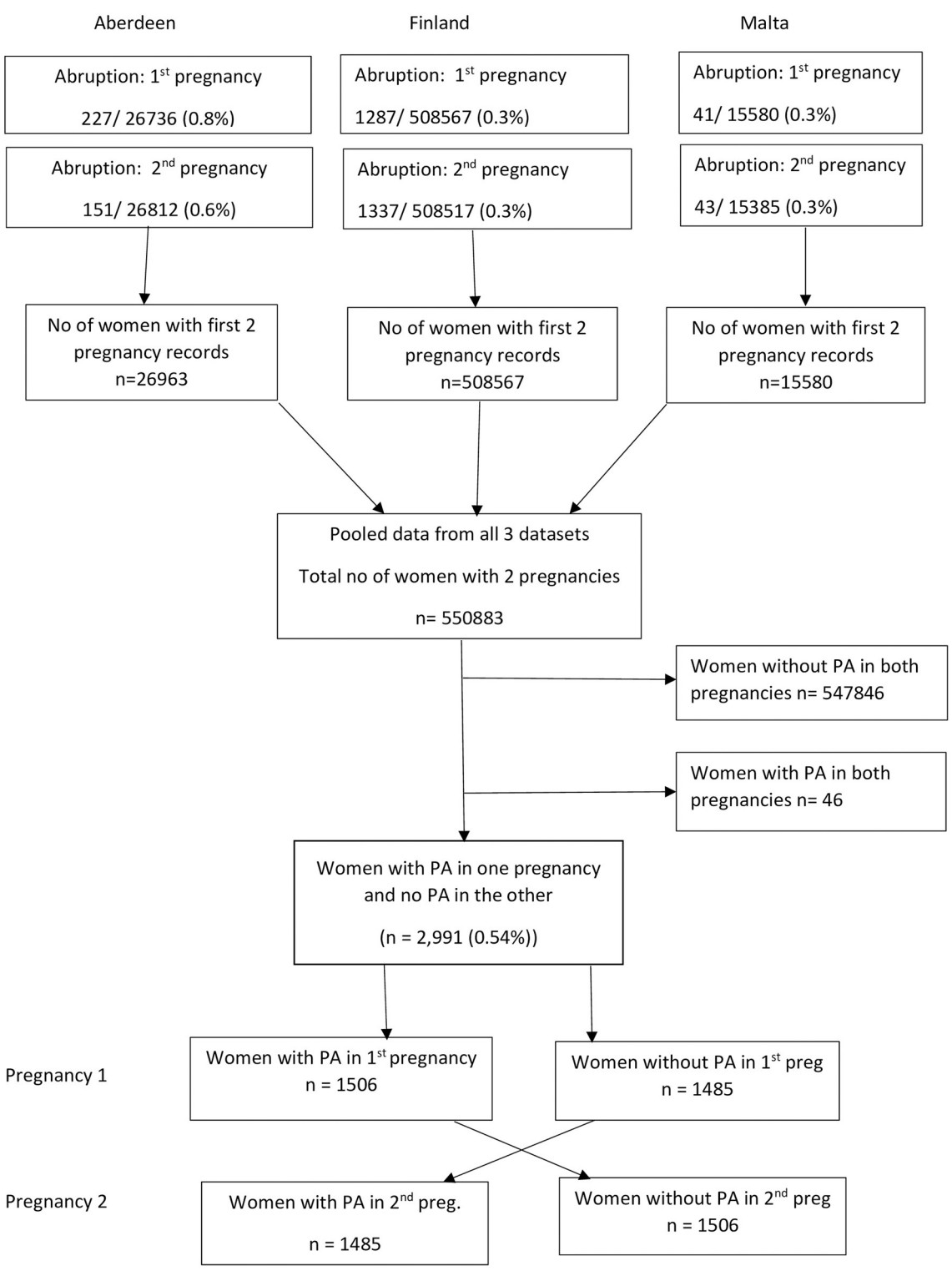

**Fig 1. Flow diagram of participant selection.**

## Statistical analysis

Datasets were cleaned and merged using IBM SPSS version 24 (Statistical Package for the Social Sciences, SPSS Inc., Chicago, IL, USA). For univariable analysis, McNemar's chi squared test of association was used to determine significant differences in the frequency of potential binary risk factors between pregnancies with and without abruption and conditional logistic regression for multinomial risk factors. Those maternal, obstetric and neonatal characteristics which were significant at p<0.2 in the univariable analysis were included in the multivariable model. However, as woman level factors such as country of origin were already matched for in the cases and controls, this was not included in the model. Multivariable conditional logistic regression with backward-selection method was used to find independent effect of risk factors on placental abruption. The strength of association was expressed as an Odds Ratio (OR) and 95% Confidence Intervals (CI). In cases where the p-value was less than 0.05 or the 95% confidence interval of the odds ratio did not include 1, the risk factor was considered to be statistically significant. Analyses were performed using Stata version 14 (StataCorp LP, College Station, TX, USA). Complete case analyses were performed after assigning a value to missing data.

## Results

Fig 1 presents cases of placental abruption by country and by pregnancy number. There were 0.8% and 0.6% cases of placental abruption in the first and second pregnancies respectively in the AMND. The incidence of placental abruption in both pregnancies was 0.3% in the Finnish and Maltese datasets. The study population comprised 2,991 women whose first two singleton pregnancies included one with PA and one without. Of these, 1,506 (50.4%) women had an abruption in pregnancy 1 and 1,485 (49.6%) experienced PA in pregnancy 2.

Tables 1 and 2 present the unadjusted (along with the counts and proportions) and adjusted models respectively investigating the association between various risk factors and placental abruption in the first and second pregnancies. Risk factors that were significantly associated with **PA in the first pregnancy** were maternal age 30–34 years {adj. OR1.35 (95% CI 1.16–1.57)} or 35–39 years {1.66 (1.31–2.12)}; smoking {1.91(1.64–2.21)}; pre-existing hypertension {1.89(1.38–2.61)}; preeclampsia {2.03(1.48–2.79)}; threatened miscarriage {2.64(1.70–4.09)}; unexplained antepartum haemorrhage {8.34(6.12–11.35)} and placenta praevia {7.26(4.71–11.19)}. After mutually adjusting for each other, risk factors that remained significantly associated with **PA in the second pregnancy** were: smoking {1.82 (1.40–2.36)}; pre-existing hypertension {2.25 (1.52–3.34)}; preeclampsia {2.61 (1.71–3.96)}; unexplained antepartum haemorrhage {9.28 (7.10–12.12)}; placenta praevia {2.70 (1.67–4.37)} and PA in the previous pregnancy {5.85 (2.84–12.04)}.

Maternal and obstetric characteristics in pregnancies with and without abruption are shown in Table 3. Results of univariable and multivariable analyses are shown in Table 4. On univariable analysis, pregnancies with abruption were significantly more likely to be associated with maternal age 35–39 years (OR 1.39; 95% CI 1.11–1.75), pre-eclampsia (OR 1.94; 95% CI 1.49–2.53), PROM (OR 1.58; 95% CI 1.11–2.25), anaemia (OR 1.66; 95% CI 1.04–2.62), threatened miscarriage (OR 1.59; 95% CI 1.20–2.11), placenta praevia (OR 4.11; 95% CI 2.49–6.78) and APH of unknown origin (OR 28.15; 95% CI 17.59–45.05) than pregnancies without abruption. Smoking status, BMI and fetal gender were not significantly associated with PA.

Results of multivariable analysis with backward elimination method for variable selection showed that maternal age 35–39 years (adjOR 1.32; 95% CI 1.01–1.73), single marital status (adjOR 1.36; 95% CI 1.04–1.76), preeclampsia (adjOR 1.69; 95% CI 1.23–2.33), APH of unknown origin (adjOR 27.05; 95% CI 16.61–44.03), placenta praevia (adjOR 3.05; 95% CI

**Table 1. Risk factors for placental abruption by pregnancy number (unadjusted analysis).**

| Characteristic | Pregnancy 1 | | | | Pregnancy 2 | | | |
|---|---|---|---|---|---|---|---|---|
| | No Abruption n(%) n = 550671 (99.7% | Abruption n (%) n = 1531(0.3%) | OR (95% CI) | P value | No abruption n(%) n = 550671 (99.7%) | Abruption n(%) n = 1531 (0.3%) | OR (95% CI) | P value |
| Maternal age in years | | | | | | | | |
| <20 | 34271 (6.2) | 131 (8.4) | **1.54 (1.27–1.86)** | **<0.001** | 4310 (0.8) | 10 (0.7) | 0.99 (0.53–1.85) | **<0.001** |
| 20–24 | 151897 (27.6) | 397 (25.5) | 1.05 (0.92–1.19) | | 76975 (14) | 162 (10.6) | 0.89 (0.75–1.08) | |
| 25–29 | 226213 (41.1) | 565 (36.3) | 1 (Ref) | | 190644 (34.6) | 447 (29.2) | 1 (Ref) | |
| 30–34 | 112125 (20.4) | 353 (22.7) | **1.27 (1.11–1.45)** | | 189396 (34.4) | 569 (37.2) | **1.28 (1.13–1.45)** | |
| 35–39 | 24365 (4.4) | 100 (6.4) | **1.65 (1.33–2.04)** | | 75559 (13.7) | 283 (18.5) | **1.59 (1.38–1.85)** | |
| >40 | 1979 (0.4) | 9 (0.6) | 1.83 (0.95–3.54) | | 13759 (2.5) | 60 (3.9) | **1.86 (1.42–2.44)** | |
| Maternal BMI | | | | | | | 1 (Ref) | |
| underweight | 7001 (1.3) | 24 (1.5) | 1.17 (0.77–1.77) | 0.462 | 7032 (1.3) | 20 (1.3) | 1.01 (0.64–1.58) | 0.968 |
| normal | 113860 (20.7) | 337 (21.7) | 1 (Ref) | | 139803 (25.4) | 394 (25.7) | 1 (Ref) | |
| overweight | 34331 (6.2) | 100 (6.4) | 0.99 (0.79–1.24) | | 53509 (9.7) | 174 (11.4) | 1.15 (0.97–1.38) | |
| obese | 15785 (2.9) | 47 (3) | 1.02 (0.75–1.38) | | 29500 (5.4) | 98 (6.4) | 1.18 (0.94–1.47) | |
| missing | 379735 (69) | 1047 (67.3) | | | 320827 (58.3) | 845 (55.2) | | |
| Single marital status | 48911 (8.9) | 192 (12.3) | **1.47 (1.26–1.71)** | **<0.001** | 35456 (6.4) | 119 (7.8) | **1.23 (1.02–1.48)** | **0.032** |
| Socially deprived | 458500 (83.3) | 1235 (79.6) | 0.99 (0.85–1.15) | 0.884 | 445845 (81) | 1212 (79.2) | 0.99 (0.86–1.13) | 0.853 |
| smoking during pregnancy | 70814 (12.9) | 348 (22.4) | **2.02 (1.79–2.27)** | **<0.001** | 61025 (11.1) | 279 (18.2) | **1.81 (1.59–2.06)** | **<0.001** |
| pre-existing hypertension | 10818 (2) | 81 (5.2) | **2.74 (2.19–3.43)** | **<0.001** | 12253 (2.2) | 71 (4.6) | **2.14 (1.68–2.72)** | **<0.001** |
| pre-existing diabetes | 2209 (0.4) | 8 (0.5) | 1.44 (0.72–2.88) | 0.309 | 4274 (0.8) | 9 (0.6) | 0.86 (0.36–2.07) | **<0.001** |
| gestational diabetes | 13229 (2.4) | 26 (1.7) | 1.26(0.76–2.13) | 0.309 | 28127 (5.1) | 97 (6.3) | **1.33 (1.08–1.64)** | **<0.001** |
| gestational hypertension | 17473 (3.2) | 79 (5.1) | **2.65 (2.14–3.28)** | **<0.001** | 12532 (2.3) | 71 (4.6) | **2.09 (1.65–2.65)** | **0.014** |
| Preeclampsia | 11975 (2.2) | 128 (8.2) | **4.04 (3.37–4.85)** | **<0.001** | 5488 (1) | 74 (4.8) | **5.05 (3.99–6.38)** | **<0.001** |
| threatened miscarriage | 8493 (1.5) | 96 (6.2) | **4.22 (3.43–5.19)** | **<0.001** | 9526 (1.7) | 67 (4.4) | **2.60 (2.03–3.32)** | 0.066 |
| APH of unknown origin | 4982 (0.9) | 295 (19) | **25.39 (22.29–28.93)** | **<0.001** | 6302 (1.1) | 288 (18.8) | **20.01 (17.56–22.81)** | **<0.001** |
| placenta praevia | 1173 (0.2) | 31 (2) | **9.57 (6.68–13.72)** | **<0.001** | 2255 (0.4) | 50 (3.3) | **8.21 (6.18–10.92)** | **<0.001** |
| anaemia in pregnancy | 4385 (0.8) | 14 (0.9) | 1.26 (0.75–2.14) | **<0.001** | 7139 (1.3) | 41 (2.7) | **2.22 (1.63–3.04)** | **<0.001** |
| Preterm Rupture of Membranes | 7705 (1.4) | 26 (1.7) | 1.19 (0.81–1.76) | **<0.001** | 9987 (1.8) | 63 (4.1) | **2.33 (1.81–2.99)** | **<0.001** |
| Male fetal gender | 268571 (48.8) | 771 (49.5) | **1.31 (1.18–1.47)** | **<0.001** | 267538 (48.6) | 738 (48.2) | 1.10 (0.99–1.22) | 0.074 |

(*Continued*)

**Table 1.** (Continued)

| Characteristic | Pregnancy 1 | | | | Pregnancy 2 | | | |
|---|---|---|---|---|---|---|---|---|
| | No Abruption n(%) n = 550671 (99.7% | Abruption n (%) n = 1531(0.3%) | OR (95% CI) | P value | No abruption n(%) n = 550671 (99.7%) | Abruption n(%) n = 1531 (0.3%) | OR (95% CI) | P value |
| Previous abruption | | | | | 1506 (0.3) | 46 (3) | **11.29 (8.39–15.21)** | **<0.001** |
| Interpregnancy interval | | | | | | | | |
| <1 yr | | | | | 5486 (1) | 31 (2) | **1.65 (1.09–2.51)** | **0.019** |
| 2 yrs | | | | | 18794 (3.4) | 77 (5) | 1.20 (0.88–1.66) | |
| 2–3 yrs | | | | | 21922 (4) | 75 (4.9) | 1 (Ref) | |
| 4 yrs | | | | | 22860 (4.2) | 86 (5.6) | 1.10 (0.81–1.57) | |
| 5 yrs | | | | | 25385 (4.6) | 69 (4.5) | 0.79 (0.57–1.10) | |
| 5–10 yrs | | | | | 148463 (27) | 323 (21.1) | **0.64 (0.49–0.82)** | |
| >10 yrs | | | | | 307856 (55.9) | 870 (56.8) | 0.83 (0.65–1.05) | |

**Table 2.** **Risk factors for placental abruption by pregnancy number (adjusted analysis).**

| Characteristic | Pregnancy 1 | | Pregnancy 2 | |
|---|---|---|---|---|
| | Adj OR (95% CI)* | P value | Adj OR (95% CI)* | P value |
| Maternal age in years | | | | |
| <20 | 0.89 (0.68–1.17) | **<0.001** | 0 | |
| 20–24 | 0.94 (0.81–1.09) | | 0.84 (0.64–1.21) | 0.42 |
| 25–29 | 1.00 | | 1.00 | |
| 30–34 | **1.35 (1.16–1.57)** | | 1.22 (0.98–1.52) | |
| 35–39 | **1.66 (1.31–2.12)** | | 1.27 (0.98–1.66) | |
| >40 | 1.78 (0.84–3.76) | | 1.40 (0.889–2.22) | |
| Deprivation | | | | |
| Not deprived | 1.00 | | 1.00 | |
| deprived | 1.18 (0.98–1.43) | 0.077 | 0.99(0.98–1.23) | 0.076 |
| smoking during pregnancy | **1.91 (1.64–2.21)** | **<0.001** | **1.82 (1.40–2.36)** | **<0.001** |
| pre-existing hypertension | **1.89 (1.38–2.61)** | **<0.001** | **2.25 (1.52–3.34)** | **<0.001** |
| gestational diabetes | 0.71 (0.45–1.06) | 0.096 | 1.29 (0.99–1.59) | 0.076 |
| gestational hypertension | 1.63 (0.93–2.28) | 0.066 | **1.59 (1.09–2.29)** | **0.014** |
| Preeclampsia | **2.03 (1.48–2.79)** | **<0.001** | **2.61 (1.71–3.96)** | **<0.001** |
| threatened miscarriage | **2.64 (1.70–4.09)** | **<0.001** | 1.70 (0.97–2.99) | 0.066 |
| APH of unknown origin | **8.34 (6.12–11.35)** | **<0.001** | **9.28 (7.10–12.12)** | **<0.001** |
| placenta praevia | **7.26 (4.71–11.19)** | **<0.001** | **2.70 (1.67–4.37)** | **<0.001** |
| Previous abruption | | | **5.85 (2.84–12.04)** | **<0.001** |

*All risk factors mutually adjusted for in the logistic regression models

Statistically significant odds ratios (95% confidence intervals) and p values are shown as bold

**Table 3. Comparison of characteristics of pregnancies with and without abruption.**

| Characteristic | | Pregnancies without abruption (n = 2991) N (%) | Pregnancies with abruption (n = 2991) N (%) | p-value |
|---|---|---|---|---|
| Maternal age in years | | | | **0.024** |
| | <20 | 117(3.9) | 138 (4.6) | |
| | 20–24 | 558 (29.6) | 542 (18.1) | |
| | 25–29 | 1046 (35.0) | 984 (32.9) | |
| | 30–34 | 886 (29.6) | 896 (30.0) | |
| | 35–39 | 323 (10.8) | 365 (12.2) | |
| | ≥40 | 61 (2.0) | 66(2.2) | |
| Pregnancy 1 | | 1485(49.6) | 1506(50.0) | 0.179 |
| Pregnancy 2 | | 1506(50.0) | 1485(49.6) | |
| Maternal BMI (Kg/m$^2$) | | | | 0.613 |
| | Underweight | 36 (1.2) | 43(1.4) | |
| | Normal weight | 690 (23.1) | 711(23.8) | |
| | Overweight | 250 (8.4) | 269 (9.0) | |
| | Obese + | 154 (5.1) | 139 (4.6) | |
| | Missing | 1861 (62.2) | 1829 (61.2) | |
| Marital status | | | | 0.327 |
| | Married/ cohabiting | 2610 (87.3) | 2574 (86.1) | |
| | Single | 284 (9.5) | 303 (10.1) | |
| | Missing | 97 (3.2) | 114 (3.8) | |
| Deprivation status | | | | 0.279 |
| | Not deprived | 437(14.6) | 449(15) | |
| | Deprived | 2418 (80.8) | 2381(79.6) | |
| | Missing | 136 (4.5) | 161(5.4) | |
| Smoking during pregnancy | | | | 0.129 |
| | No | 2309(77.2) | 2279 (76.2) | |
| | Yes | 602 (20.1) | 605 (20.2) | |
| | Missing | 80 (2.7) | 107 (3.6) | |
| Gestational Hypertension | | | | 0.763 |
| | No | 2843 (95.1) | 2849 (95.3) | |
| | Yes | 148 (4.9) | 148 (4.9) | |
| Preeclampsia | | | | **<0.001** |
| | No | 2876 (96.2) | 2797 (93.5) | |
| | Yes | 115 (3.8) | 194 (6.5) | |
| Gestational Diabetes | | | | 0.255 |
| | No | 2533(84.7) | 2509 (83.9) | |
| | Yes | 98(3.3) | 122 (4.1) | |
| | Missing | 360 (12.0) | 360 (12.0) | |
| Threatened Miscarriage | | | | **0.004** |
| | No | 2883 (96.4) | 2836 (94.8) | |
| | Yes | 108 (3.6) | 155 (5.2) | |
| APH of unknown origin | | | | **<0.001** |
| | No | 2922 (97.7) | 2435 (81.4) | |
| | Yes | 69 (2.3) | 556 (18.6) | |
| Placenta Praevia | | | | **<0.001** |
| | No | 2970 (99.3) | 2911 (97.3) | |
| | Yes | 21 (0.7) | 80 (2.7) | |

*(Continued)*

**Table 3.** (Continued)

| Characteristic | Pregnancies without abruption (n = 2991) N (%) | Pregnancies with abruption (n = 2991) N (%) | p-value |
|---|---|---|---|
| Anaemia in pregnancy | | | 0.133 |
| No | 2518 (84.2) | 2499 (83.6) | |
| Yes | 36 (1.2) | 55 (1.8) | |

1.74–5.36) were more likely to be independently, significantly associated with pregnancies with abruption. Maternal anemia, threatened miscarriage and PROM, which were significantly associated with abruption at univariable analysis were no longer statistically significant in the multivariable model.

Although this was not the primary focus of this study, the perinatal outcomes of pregnancies with and without placental abruption are presented as S1, S2 and S3 Tables. In both pregnancies and in unadjusted as well as multi-adjusted models, placental abruption was significantly associated with Caesarean or instrumental delivery, stillbirth, preterm birth, low birth weight and IUGR.

**Table 4. Unadjusted and adjusted odds ratios (95% Confidence Intervals) for case crossover analysis.**

| Characteristic | Unadjusted OR (95% CI) | Adjusted OR (95%CI)* |
|---|---|---|
| Maternal age in years | | |
| <20 | 1.31 (0.95–1.81) | 1.06 (0.72–1.57) |
| 20–24 | 1.03 (0.86–1.23) | 0.99 (0.80–1.21) |
| 25–29 | 1.00 | 1.00 |
| 30–34 | 1.14 (0.99–1.32) | 1.13 (0.96–1.33) |
| 35–39 | **1.39 (1.11–1.75)** | **1.32 (1.01–1.73)** |
| ≥40 | 1.45 (0.93–2.27) | 1.06 (0.64–1.78) |
| maternal BMI | | |
| Underweight | 1.38 (0.71–2.67) | 1.59 (0.69–3.64) |
| Normal | 1.00 | 1.00 |
| Overweight | 1.00 (0.77–1.32) | 0.95 (0.68–1.33) |
| Obese | 0.72 (0.49–1.08) | 0.76 (0.46–1.26) |
| Single Marital Status | 1.14 (0.92–1.42) | **1.36 (1.04–1.76)** |
| Deprived | 0.83 (0.63–1.09) | 0.79 (0.57–1.10) |
| Smoking During Pregnancy | 1.09 (0.86–1.38) | 1.06 (0.81–1.38) |
| Gestational Diabetes | 1.34 (0.98–1.82) | 1.25 (0.89–1.77) |
| Gestational Hypertension | 0.95 (0.74–1.22) | 0.99 (0.71–1.37) |
| Preeclampsia | **1.94 (1.49–2.53)** | **1.69 (1.23–2.33)** |
| Threatened Miscarriage | **1.59 (1.20–2.11)** | 1.32 (0.88–1.97) |
| APH of Unknown Origin | **28.15 (17.59–45.05)** | **27.05 (16.61–44.03)** |
| Placenta Praevia | **4.11 (2.48–6.78)** | **3.05 (1.74–5.36)** |
| Anaemia In Pregnancy | **1.66 (1.04–2.62)** | 1.43 (0.87–2.35) |
| Preterm Prelabour Rupture of Membranes | **1.58 (1.11–2.25)** | 1.38 (0.92–2.08) |
| Male Fetal Gender | 1.03 (0.93–1.15) | 1.01 (0.90–1.14) |

Statistically significant odds ratios are shown as bold.

*Adjusted for all other variables in the model

## Discussion

### Main findings

In a pooled dataset from three European populations comparing pregnancies occurring in the same woman we found that pregnancies with abruption were more likely to be associated with pre-eclampsia, placenta praevia and APH of unknown origin. Abruption was also more likely to occur in older women and those who were single. PPROM, threatened miscarriage and maternal anaemia were not confirmed as significant risk factors for PA in the multivariable model. Smoking status, BMI and fetal gender were not significantly associated with PA in univariable or multivariable models.

### Comparison with existing literature

Our finding that pre-eclampsia increased the odds of abruption is supported by previous studies. Kramer found an odds ratio of 2.05 [18] and Lindqvist and Happach reported a 3.4-fold increased risk. [11]

Abruption was associated with maternal age 35–39 years—a finding which is consistent with the existing literature linking maternal age ≥35 years with PA with adjusted OR of 1.62 [19] We found no association with age ≥40 years but this is likely due to the small number of women in this group. The association between increased maternal age and abruption is suggested to be due to decreased vascularisation of the uterus which occurs with age and predisposes to placental insufficiency. [1] While other studies have also found a link between decreased maternal age (<20 years) and abruption, [20] this study found no evidence supporting this.

Kramer [18] also found that single marital status was associated with an increased risk of placental abruption and their odds ratio of 1.50 (95% CI 1.13–1.98) is comparable to our findings of 1.36 (95% CI 1.04–1.76).

It is notable that we did not find maternal smoking or BMI to be associated with abruption, this is probably explained by the fact that smoking status and BMI did not often change between successive pregnancies.

APH of unknown origin and placenta praevia were associated with pregnancies with placental abruption. Baumann [21] found the risk of abruption from bleeding >28 weeks' gestation (adj OR 18.7 95% CI 14.2–24.6) and placenta praevia (adj OR 4.3; 95% CI 2.7–6.9) to be of a similar magnitude to the risk from APH of unknown origin and placenta praevia found in this study. Baumann admitted that they did not know in how many instances the APH coincided with the index abruption, thus having no predictive value. APH of unknown origin could be an early indicator or sign of placental abruption rather than a risk factor per se. An association with vaginal bleeding in early pregnancy (<27 weeks) was identified by Ananth, [5] who found it to increase risk of PA (adjusted relative risk 3.1; 95% CI 2.3, 4.1). They argued that this was a risk factor and not an early predictor due to the low positive predictive value of vaginal bleeding for placental abruption (3%), but high negative predictive value (98%), and that these results support the theory that PA is the result of chronic placental pathology beginning early in pregnancy—which manifests as abnormal bleeding.

### Strengths and limitations

A major strength of this study is the novel use of a case-crossover design to compare risk factor exposure between pregnancies with and without PA. The existing literature consists of cohort and case-control analyses; this design adds a different and complementary perspective where women act as their own controls, thereby minimising within woman confounding. Similar

results to previous studies verify the value of this design. The case cross-over study design is a relatively new epidemiological method that is a variation of a case control study and is self-matched. [22] It allows the study of transient exposures on an acute and rare outcome, in this case placental abruption. [23] This allowed us to examine the effect of risk factors such as age, pre-eclampsia and smoking status that may alter between pregnancies. Self-matching of cases reduces control-selection bias [23] and means that women act as their own control; the pregnancy with abruption is the case and the pregnancy without abruption is the control, with the PA/unaffected pregnancy in either order. Furthermore, self-matching removed the effect of genetic factors, which are known to play a role, [12] and other unmeasured confounding. This allowed reliable examination of the impact of transient clinical and socio-demographic risk factors such as age, fetal gender smoking and hypertension. The aim of the study was to look at the effect of changing some of the already known risk factors on the occurrence of placental abruption keeping the woman-based factors (eg. Genetic predisposition) constant. In fact our starting point was to identify the risk factors implicated in the literature for abruption and see what difference any change in these would make.

Further strengths of this study are related to the size and quality of the datasets. Pooling data from three sources provided a relatively large study population, thus allowing us to explore an uncommon condition. The databases used are reliable and well-established and contain information on complete populations of women for a long period of time and the data are recent, up to 2015. The detailed information allowed a comprehensive study of many potential risk factors in relation to PA.

While two data sources (Maltese NOIS and Finnish MBR) capture national data, AMND contains data gathered from Aberdeen Maternity Hospital (AMH) which is the only hospital to serve the entire population of the region (Aberdeen City District) which offers no other maternity facilities, either private or public. This represents two potential limitations. First, Aberdeen is a relatively affluent area which may not be representative of the total Scottish population. Second, as a tertiary referral centre, Aberdeen Maternity Hospital receives a disproportionate number of more complicated cases from outside the region. This is confirmed by the increased prevalence of PA seen in the Aberdeen data compared to Maltese and Finnish data. However, as only women who had a pregnancy with placental abruption were included in this study, this is unlikely to have a major effect on the findings.

Large amounts of missing or unrecorded data for some variables meant that substance misuse, alcohol use and in-vitro fertilisation (IVF) conception could not be included as co-variates. Tests of association would be weak with >60% missing data, and for drug and alcohol use self-reporting is likely to produce underestimations. [18] This can partially be attributed to some variables not being recorded in all three sources; drug and alcohol use was only recorded in the Maltese data. Excluding these variables meant that their effects could not be investigated and their unobserved effects could act as residual confounding.

COS are defined sets of outcomes relevant to a particular condition or topic, developed by the Core Research Outcomes in Women's and Newborn Heath Initiative (CROWN). This initiative is in response to heterogeneity in the outcomes investigated by studies looking at the same problem. This variation limits their comparison and leads to outcome reporting bias and difficulty or inaccuracy in systematic reviews. [24] This lack of clarity is likely to hinder or delay the implementation of research findings into clinical practice. Additionally, the definition of covariates differed in the three datasets. For example, socioeconomic status was based on the mother's occupation in the Finnish data, maternal education level in the Maltese data while post code based deprivation category was used in the AMND. Consequently, we had to arbitrarily categorise all data as 'deprived' and 'non-deprived' for consistency. The data although spanning three European countries, are derived from a mainly white Caucasian

population and therefore may not be generalisable to other populations with different health care systems and access.

## Interpretation

While a number of studies have previously established the presence and magnitude of risk of factors such as pre-eclampsia, few previous studies have investigated APH of unknown origin and placenta praevia as risk factors for placental abruption. This may be related to way data on APH is coded in registries—some have a hierarchical coding system whereby it is impossible for placental abruption and placenta praevia to be coded as comorbidities. In the Medical Birth Register in Finland there are two check boxes, and the birth hospitals report these diagnoses at the same time: Placenta previa; and Ablatio placentae (premature detachment of placenta) only if diagnosed during delivery. We looked at the overlap in these diagnoses and on average 5–6 cases were diagnosed as both over the years. Thus although the absolute numbers were small, the relative risk was high. While these results should be interpreted with caution, the size of the risk they confer in this study is substantial and warrants further investigation. These results suggest that clinicians should be aware that any unexplained bleeding or diagnosis of placenta praevia could mean that women are at a much higher risk of abruption later in pregnancy. In the U.K., a recent Royal College of Obstetricians and Gynaecologists guideline (2018) advises earlier planned delivery with confirmed placenta praevia at 36–37 weeks as the risk of increased bleeding and the need for emergency delivery increases with advancing gestation. [25] This risk increases rapidly after 36 weeks of gestation; below 5% by 35 weeks, 15% by 36 weeks, 30% by 37 weeks and 59% by 38 weeks of gestation. [25]. It could be argued that increased risk of bleeding could be in part due to a higher risk of PA in these women. In addition, the 2011 RCOG guidelines on antepartum haemorrhage state that following APH of unknown origin the pregnancy should be re-classified as high risk of PA; [2] this is in keeping with the results of this study.

Of all the risk factors that were found to be independently associated with PA in this study, advanced maternal age was the only one that was potentially modifiable. This increased risk was independent of parity, signifying that not only first pregnancies but also subsequent pregnancies were at higher risk of complications if occurring in women aged 35 and over. The UK Office for National Statistics (ONS) recently published data showing that in 2017 fertility rates decreased for every age group, except for women over 40 which increased by 1.6%. [26] Older women are making up an increased proportion of obstetric patients. Advanced maternal age comes with a spectrum of increased clinical risk; both maternal complications such as pre-eclampsia, gestational diabetes, placental abruption and adverse perinatal outcomes including preterm birth, miscarriage, stillbirth, growth restriction and genetic disorders. [27] This association is suggested to be due to placental dysfunction. Targeted public health messages should advise women of the higher risks associated with conceiving over the age of 35. Those planning to conceive a second time should also be advised not to wait too long as this study showed that advanced maternal age even in the second pregnancy conferred an increased risk of PA and other placental dysfunction.

## Conclusion

Risk factors for PA include APH of unknown origin, placenta praevia, pre-eclampsia, maternal age ≥35 and single marital status. Women with APH of unknown origin and placenta praevia should be classified as at high risk for PA. Our data confirms preeclampsia as a well-established risk factor for PA. Knowledge that PA is more common in older women could help to inform clinical decision making in pregnancy.

## Supporting information

**S1 Checklist. STROBE statement—Checklist of items that should be included in reports of observational studies.**
(DOC)

**S1 Table. Comparison of perinatal outcomes of the first pregnancy with and without placental abruption.**
(DOCX)

**S2 Table. Comparison of perinatal outcomes of 2nd pregnancy with and without placental abruption.**
(DOCX)

**S3 Table. Unadjusted and adjusted odds ratios (95% Confidence Intervals) of perinatal outcomes in pregnancies 1 and 2.**
(DOCX)

## Acknowledgments

With thanks to the women from Malta, Finland and Aberdeen whose data is included and to all who contributed to the datasets used.

## Author Contributions

**Conceptualization:** Ashalatha Shetty, Siladitya Bhattacharya, Sohinee Bhattacharya.

**Data curation:** Mika Gissler, Miriam Gatt, Sohinee Bhattacharya.

**Formal analysis:** Emma Anderson, Edwin Amalraj Raja.

**Methodology:** Edwin Amalraj Raja.

**Supervision:** Edwin Amalraj Raja, Ashalatha Shetty, Siladitya Bhattacharya, Sohinee Bhattacharya.

**Writing – original draft:** Emma Anderson.

**Writing – review & editing:** Edwin Amalraj Raja, Ashalatha Shetty, Mika Gissler, Miriam Gatt, Siladitya Bhattacharya, Sohinee Bhattacharya.

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
