## [Decision Letter · Decision Letter 0]

4 Mar 2020

PONE-D-20-03397

Changing risk factors for placental abruption: a case crossover study using routinely collected data from Finland, Malta and Aberdeen

PLOS ONE

Dear Dr Bhattacharya,

Thank you for submitting your manuscript to PLOS ONE. After careful consideration, we feel that it has merit but does not fully meet PLOS ONE’s publication criteria as it currently stands. Therefore, we invite you to submit a revised version of the manuscript that addresses ALL the points raised during the review process.

SPECIFIC ACADEMIC EDITOR COMMENTS: Your manuscript was reviewed by two experts in the field. Although interest was found in your study, there were several major comments and questions that arose during the review process. These include, but are not limited to the need to clarify the novelty of this specific study, because the majority of these findings are already known. Furthermore, there were requests for additional data about birth weight, IUGR, week of delivery, mode of delivery, patients with induced labor, and specifics about IVF treatment and alcohol use. And the question was raised about how placental abruption was diagnosed in the cases of placenta previa.

We would appreciate receiving your revised manuscript by Apr 18 2020 11:59PM. To enhance the reproducibility of your results, we recommend that if applicable you deposit your laboratory protocols in protocols.io, where a protocol can be assigned its own identifier (DOI) such that it can be cited independently in the future. For instructions see: http://journals.plos.org/plosone/s/submission-guidelines#loc-laboratory-protocols

We look forward to receiving your revised manuscript.

Kind regards,

Frank T. Spradley

Academic Editor

PLOS ONE

Journal Requirements:

3. Thank you for stating the following financial disclosure: "No"

a)    Please provide an amended Funding Statement that declares *all* the funding or sources of support received during this specific study (whether external or internal to your organization) as detailed online in our guide for authors at http://journals.plos.org/plosone/s/submit-now.  

b)    Please state what role the funders took in the study.  If any authors received a salary from any of your funders, please state which authors and which funder. If the funders had no role, please state: "The funders had no role in study design, data collection and analysis, decision to publish, or preparation of the manuscript."

4. Thank you for stating the following in your Competing Interests section:  "No"

Reviewers' comments:

Reviewer's Responses to Questions

**Comments to the Author**

1. Is the manuscript technically sound, and do the data support the conclusions?

Reviewer #1: Yes

Reviewer #2: Yes

2. Has the statistical analysis been performed appropriately and rigorously? 

Reviewer #1: Yes

Reviewer #2: Yes

3. Have the authors made all data underlying the findings in their manuscript fully available?

Reviewer #1: No

Reviewer #2: Yes

4. Is the manuscript presented in an intelligible fashion and written in standard English?

Reviewer #1: Yes

Reviewer #2: Yes

5. Review Comments to the Author

Reviewer #1: Changing risk factors for placental abruption: a case crossover study using routinely collected data from Finland, Malta and Aberdeen.

This study is a case-crossover study design with records of the two first pregnancies from women who had PA in one pregnancy and not the other.

Cases were pregnancies with abruption and matched controls were pregnancies without abruption in the same woman. A total of 2991 women were included in their study.

The study is well written, and I command the authors for the study design and size of the cohort.

However, I have concerns about the importance and novelty of the data since all risk factors statistically significant in the study are well known in the literature.

Furthermore, in my opinion, some important data is lacking which could influence the results and for example: *Week of delivery, Induction of labor

*Mode of delivery- mainly important in the second pregnancy after PA. Since previous PA is a strong and well known it is possible that an elective Cesarean delivery was performed in an earlier week and affected the occurrence of PA in the coming pregnancy. I ask the authors to add this data if possible.

* IVF treatment and alcohol use- As the authors mentioned in the limitations of the study.

Regarding placenta previa as a risk factor for placenta abruption, how was the diagnosis of abruption done in the cases of placenta previa? Since bleeding is a common symptom did data regarding placenta pathology is present in the registry?

Reviewer #2: I really enjoyed reading your paper. It is refreshing, well designed and well written. I have some minor comments:

1. Is it any reason that you did not include birth weight or more specifically intrauterine growth restriction (IUGR) in your data?

2. Could you please provide the numbers and percentages in your tables as well as OR, CI and P-values?

3. I think you should categorize maternal age into only 3 categories: <20, 20-35 and >35. This is only a suggestion; you don’t have to do that.

6. PLOS authors have the option to publish the peer review history of their article (what does this mean?). If published, this will include your full peer review and any attached files.

Reviewer #1: No

Reviewer #2: Yes: Elham Baghestan

---

## [Author Response · Author response to Decision Letter 0]

30 Apr 2020

Reviewer #1:

This study is a case-crossover study design with records of the two first pregnancies from women who had PA in one pregnancy and not the other. Cases were pregnancies with abruption and matched controls were pregnancies without abruption in the same woman. A total of 2991 women were included in their study. The study is well written, and I command the authors for the study design and size of the cohort.

We thank the reviewer for their supportive comments.

However, I have concerns about the importance and novelty of the data since all risk factors statistically significant in the study are well known in the literature.

The aim of the study was to look at the effect of changing some of the already known risk factors on the occurrence of placental abruption keeping the woman-based factors (eg. Genetic predisposition) constant. In fact our starting point was to identify the risk factors implicated in the literature for abruption and see what difference any change in these would make. The novelty of this analysis lies in studying the change in these risk factors and not the risk factors per se. We have added a few sentences in the discussion to reflect this.

Furthermore, in my opinion, some important data is lacking which could influence the results and for example: *Week of delivery, Induction of labor,*Mode of delivery- mainly important in the second pregnancy after PA. Since previous PA is a strong and well known it is possible that an elective Cesarean delivery was performed in an earlier week and affected the occurrence of PA in the coming pregnancy. I ask the authors to add this data if possible; * IVF treatment and alcohol use- As the authors mentioned in the limitations of the study.

Thank you for these suggestions. As we were studying the risk factors preceding placental abruption we did not include the outcomes of pregnancies affected by abruption. We have these outcome data available apart from induction of labour which has a lot of missing data and have added these in additional supplementary tables in the manuscript. We did not have data relating to IVF treatment and alcohol use in all the constituting datasets and have already mentioned this as a weakness in the discussion. 

Regarding placenta previa as a risk factor for placenta abruption, how was the diagnosis of abruption done in the cases of placenta previa? Since bleeding is a common symptom did data regarding placenta pathology is present in the registry?

We looked into this further and found that this was driven by the way data was collected in the Finnish birth registry. In the Medical Birth Register in Finland there are two check boxes, and the birth hospitals report these diagnoses at the same time: Placenta previa; only if diagnosed during delivery and Ablatio placentae (premature detachment of placenta). We looked at the overlap in these diagnoses and on average 5 -6 cases were diagnosed as both in each year of delivery. Thus although the absolute numbers were small, the relative risk was high.

Reviewer #2: I really enjoyed reading your paper. It is refreshing, well designed and well written.

Many thanks for the supportive comments. 

I have some minor comments:

1. Is it any reason that you did not include birth weight or more specifically intrauterine growth restriction (IUGR) in your data?

As stated earlier, we were interested in looking at the risk factors for placental abruption and not the outcomes of the pregnancy and therefore did not include birthweight or IUGR in the manuscript. We have now included supplementary tables with the outcomes of pregnancy.

2. Could you please provide the numbers and percentages in your tables as well as OR, CI and P-values?

Thank you for this suggestion and we apologise for this oversight. This has now been included.

3. I think you should categorize maternal age into only 3 categories: <20, 20-35 and >35. This is only a suggestion; you don’t have to do that.

Many thanks for this suggestion but we thought there would be more granularity in the data if we categorised maternal age into smaller categories and the relatively large combined dataset allowed us to do so. 

Editorial comment:

If there are ethical or legal restrictions on sharing a de-identified data set, please explain them in detail (e.g., data contain potentially identifying or sensitive patient information) and who has imposed them (e.g., an ethics committee). Please also provide contact information for a data access committee, ethics committee, or other institutional body to which data requests may be sent.

The dataset was created from three population based international datasets and permissions obtained from governing committees for the 3 databases. Therefore permission for public access to data will need to be given by all three committees. The Finnish register data have been given for this specific study, and the data cannot be shared without authorization from the register keepers. More information on the authorization application to researchers who meet the criteria for access to confidential data can be found at 

https://thl.fi/fi/web/thlfi-en/statistics/information-for-researchers/authorisation-application (THL). 

Similarly data from Aberdeen can be accessed by applying to the AMND steering committee found at https://www.abdn.ac.uk/iahs/research/obsgynae/amnd/access.php

The data will be deposited in a University of Aberdeen repository after the manuscript is accepted for publication.

---

## [Decision Letter · Decision Letter 1]

11 May 2020

Changing risk factors for placental abruption: a case crossover study using routinely collected data from Finland, Malta and Aberdeen

PONE-D-20-03397R1

Dear Dr. Bhattacharya,

We are pleased to inform you that your manuscript has been judged scientifically suitable for publication and will be formally accepted for publication once it complies with all outstanding technical requirements.

With kind regards,

Frank T. Spradley

Academic Editor

PLOS ONE

Reviewers' comments:

Reviewer's Responses to Questions

**Comments to the Author**

1. If the authors have adequately addressed your comments raised in a previous round of review and you feel that this manuscript is now acceptable for publication, you may indicate that here to bypass the “Comments to the Author” section, enter your conflict of interest statement in the “Confidential to Editor” section, and submit your "Accept" recommendation.

Reviewer #2: All comments have been addressed

2. Is the manuscript technically sound, and do the data support the conclusions?

Reviewer #2: Yes

3. Has the statistical analysis been performed appropriately and rigorously? 

Reviewer #2: Yes

4. Have the authors made all data underlying the findings in their manuscript fully available?

Reviewer #2: Yes

5. Is the manuscript presented in an intelligible fashion and written in standard English?

Reviewer #2: Yes

6. Review Comments to the Author

Reviewer #2: The autors have responded adequately to my comments. I have no further comments and wish you luck with the publication.

7. PLOS authors have the option to publish the peer review history of their article (what does this mean?). If published, this will include your full peer review and any attached files.

Reviewer #2: Yes: Elham Baghestan

---

## [Editor Report · Acceptance letter]

22 May 2020

PONE-D-20-03397R1 

Changing risk factors for placental abruption: a case crossover study using routinely collected data from Finland, Malta and Aberdeen 

Dear Dr. Bhattacharya:

I am pleased to inform you that your manuscript has been deemed suitable for publication in PLOS ONE. Congratulations! Your manuscript is now with our production department. 

With kind regards,

on behalf of

Dr. Frank T. Spradley 

Academic Editor

PLOS ONE